# Mechanical Aging Test and Sealing Performance of Thermoplastic Vulcanizate as Sealing Gasket in Automotive Fuel Cell Applications

**DOI:** 10.3390/polym15081872

**Published:** 2023-04-13

**Authors:** Hyungu Im, Sunkyoung Jeoung

**Affiliations:** Material Technology R&D Division, Korea Automotive Technology Institute, Cheonan-si 31214, Republic of Korea

**Keywords:** mechanical ageing, sealing performance, fuel cell, thermoplastic vulcanizate, EPDM/TPV

## Abstract

Ethylene–propylene–diene monomer (EPDM) rubber is one of the rapidly developing synthetic rubbers for use as a gasket material in proton exchange membrane (PEM) fuel cell applications. Despite its excellent elastic and sealing properties, EPDM faces challenges such as molding processability and recycling ability. To overcome these challenges, thermoplastic vulcanizate (TPV), which comprises vulcanized EPDM in polypropylene matrix, was investigated as a gasket material for PEM fuel cell applications. TPV showed better long-term stability in terms of tension and compression set behaviors under accelerated aging conditions than EPDM. Additionally, TPV exhibited significantly higher crosslinking density and surface hardness than EPDM, regardless of the test temperature and aging time. TPV and EPDM showed similar leakage rates for the entire range of test inlet pressure values, regardless of the applied temperature. Therefore, we can conclude that TPV exhibits a similar sealing capability with more stable mechanical properties compared with commercialized EPDM gaskets in terms of He leakage performance.

## 1. Introduction

Various renewable and clean energy sources have been investigated and applied at various levels. However, the unique intermittence and instability of renewable energy pose major challenges to the stable operation of power systems, opening temporal and spatial gaps between energy consumption by end-users and its availability. Thus, hydrogen energy based on electrochemical energy storage technology has received significant attention to address the recent renewable energy challenge owing to its high energy density, environmentally friendly process, and relatively abundant sources [1,2,3,4,5]. Many countries have already implemented hydrogen energy development strategies. Proton exchange membrane (PEM) fuel cells, which are highly efficient hydrogen conversion devices, can play an important role in various applications, including transportation, portable devices, and backup power systems, owing to their high power density and simplicity of design. The low open-circuit voltage of PEM fuel cells is attributed to several structural factors, such as H_2_ crossover and contamination; therefore, many unit cells are usually assembled in series to construct a stacked structure to achieve the high power and voltage required in electrical devices and automotive applications [6].

In fuel cells, hydrogen, oxygen, and the coolant should be isolated in their respective channels by sealing gaskets. Sealing failure results in the leakage, mixing, and reaction of the reactant gases, causing local hot-spots and the final failure of the stack [7]. Hence, during the assembly process, sealing gaskets made of polymeric materials are placed and compressed on both sides of the bipolar plate to prevent the fluids from mixing. Stable sealing is necessary for the safe operation of fuel cells; gasket delamination and slippage, resulting from insufficient pressure and excessive pressure, respectively, are common modes of sealing failure.

The gasket, together with its sealing structure, is a key factor governing not only the performance and durability but also the safety of fuel cells [8]. Gaskets must combine different properties, such as electrical insulation, gas prevention, and chemical resistance, to allow oxidant gas to be used and high mechanical resistance to withstand compression in the stack [9]. In general, it is difficult to simultaneously meet all the requirements of gaskets. Thus, the proper choice of sealing material should be based on the chemical and physical properties required for the specific operating conditions.

Many elastic materials, such as copolymeric resin (CR), liquid silicone rubber (LSR), fluorosilicone rubber (FSR), ethylene–propylene–diene monomer rubber (EPDM), and fluorocarbon elastomer/fluorocarbon monomer (FKM/FPM), have been investigated for use as gasket materials with proven performance based on required performance indicators. Some commonly used gasket materials for low-temperature fuel cells include silicon, PTFE, and EPDM rubber, which are ostensibly the best materials for ensuring gas tightness [10,11].

Among these candidates, EPDM rubber is one of the most rapidly developing synthetic rubbers and possesses inherent resistance to heat, light, and fuel/oil [12,13,14] owing to its saturated hydrocarbon backbone with the presence of double bonds in the side chains. Thus, it is widely used as a seal or gasket material in many industrial applications, including fuel cells [15,16,17]. Moreover, in the recently commercialized fuel cell electric vehicle, EPDM has been adopted as the sealing material in the fuel cell stack that prevents hydrogen, oxygen, and water from leaking within the plate-shaped separator and the power components of the fuel cell. Despite its excellent merits, EPDM requires improvement in terms of its recycling ability and manufacturing processability.

For instance, the direct injection molding of EPDM is performed on the sealing portion of a bipolar plate, and the EPDM–bipolar plate assembly is exposed to high ambient temperatures for the vulcanization of EPDM. The scrap material in this process exhibits misalignment, large thickness deviation, and other process defects resulting from limited process capability during manufacturing, as well as limited recycling ability owing to the irreversible processability of vulcanized elastomer materials [18]. Therefore, if a gasket material exhibits a recycling ability with a consistent sealing property, similar to elastomer gasket materials, it represents a good opportunity for the PEM fuel cell industry to overcome the challenges of process complexities and increased cost.

In this study, thermoplastic vulcanizate (TPV) was investigated as a candidate material for use as gaskets in automotive PEM fuel cell applications as an alternative to EPDM, because it exhibits the properties of both rubber and thermoplastics, i.e., it possesses the good elasticity and mechanical properties of thermoset rubbers, as well as the good processability and recycling ability of thermoplastics. Indeed, it can be processed using classical melt processing techniques, such as extrusion, injection molding, compression molding, and blow molding, and its production scrap can be easily recycled [19,20,21,22,23]. Therefore, many practices exist for the application of TPV as a sealing material in various applications, such as exterior door and window sealing. However, despite its good processability and cost merits, few reports are available on the use of TPV as a fuel cell gasket material based on its chemical resistance in the electrochemical environment.

This study aims to understand the mechanical stability of TPV as a gasket material when exposed to the PEM fuel cell environment, such as high temperatures and acidic conditions, and determine whether it can replace the commercialized EPDM gasket material in fuel cell applications. Therefore, mechanical property changes, including tension and compression set (CS) behaviors, hardness changes, and physical free volume changes, were investigated under accelerated aging conditions. Finally, the sealing performance of each material was investigated and compared for use as a gasket material in automotive PEM fuel cell applications.

## 2. Experimental Section

### 2.1. Material

EPDM (KEP 9520; 56.5 wt.% ethylene and 9.0 wt.% 5-ethylidene-2-norbornene (ENB)) was purchased from Kumho Polychem (Seoul, Republic of Korea). 1,2-Butadiene was supplied by JSR Corporation (Tokyo, Japan). Other rubber ingredients were industrially available products and were used as received. The EPDM rubber gaskets used in this study were compounded by Hwaseung Material (Yangsan, Republic of Korea).

A polypropylene (PP) homopolymer (HD120MO) was obtained from Korea Petrochemical Ind. Co., Ltd., (Seoul, Republic of Korea). To prepare TPV, which is a blended material composed of crosslinked EPDM (KEP 2480; 57.5 wt.% ethylene and 8.9 wt.% 5-ethylidene-2-norbornene (ENB); Kumho Polychem, Seoul, Republic of Korea) in PP matrix, the EPDM phase was first introduced into the blender and sheared for 2 min at 120 rpm and 200 °C. The plasticizer was then poured in, and the blend was mixed until torque stabilization. Immediately after torque stabilization, PP was introduced, and the torque was again left to stabilize. For reactive blends, the curing system (2,5-dimethyl-2,5-di-(tert-butylperoxy)hexane; Teahwa Cooperation, Republic of Korea) was added, and mixing was continued for 10 min to ensure the complete crosslinking of EPDM.

Finally, the blends were compression-molded at 200 °C for 2 min into 2 mm thick samples that were stored away from light and heat until testing.

### 2.2. Mechanical Properties

The tensile property of rubber gaskets was measured using a 5567 universal testing machine by Instron Co. (Norwood, MA, USA) by following the ISO 37-2005 standard. Pristine and aged gaskets were cut into dumbbell-shaped thin strips with dimensions of 75 mm × 4 mm × 2 mm. The tensile test was conducted at a speed of 500 mm min^−1^, and the average value of five samples was taken.

The CS test of pristine and aged rubber gaskets was conducted during the compressive stress–thermo-oxidative aging process, which involved the periodic removal of the compressive jigs from the ovens and cooling for 2 h at room temperature. The CS was calculated according to the following equation:(1)$$C=\frac{h_{0}-h_{t}}{h_{0}-h_{s}}\times 100\%$$
where h_0_ and h_t_ are the gasket heights before and after aging, respectively, and h_s_ is the gasket height compressed to a certain joint opening displacement (JOD). A CS of 0% implies a complete recovery of the initial height, while a CS of 100% implies a complete loss of elasticity.

### 2.3. Crosslinking Density

The crosslinking density (Ve) was determined using the equilibrium swelling method. The pristine and aged gasket material samples were immersed in cyclohexane for one week, during which the solvent was replaced with fresh cyclohexane every three days. We measured the crosslinking density of the rubber samples according to the ASTM D6814 standard. The rubber samples were immersed in solvent dabbed with paper towels and then weighed.

### 2.4. Hardness

The Shore A hardness of pristine and aged gasket material samples was measured using a Shore hardness tester, TH200 (Shanghai Shuangxu Electronics Co., Ltd., Shanghai, China), according to the ISO 868 standard. A minimum of five tests were performed on each sample, and the average was taken.

### 2.5. Fourier-Transform Infrared (FTIR) Spectroscopy

The chemical structure analysis of EPDM and TPV with thermal aging was conducted using a Nicolet-560 Fourier-transform infrared (FTIR) spectrometer (Norwood, MA, USA) in attenuated total reflectance mode from 4000 to 700 cm^−1^; a series of 32 scans and resolution of 1 cm^−1^ were employed.

### 2.6. Sealing Property

TPV and EPDM were injection-molded according to the recommended profile provided by the supplier to obtain a desired shape. The EPDM material sheet was prepared with the open roll compression process at an operating temperature of 190 °C and was cut to the required dimensions, i.e., approximately 2 mm in thickness and 80 cm in circumference. Finally, the prepared material was vulcanized using a preheated mold at 200 °C for 2 min. The TPV gasket material was prepared using injection molding under a 300-ton force and at a molding temperature of 200 °C.

Furthermore, to simulate the stack operation environment of the fuel cell, a fabricated gasket sample and a metal separator (SUS304) were alternately laminated four times and then compressed to 40% of the initial gasket thickness. He (99.9%), instead of H_2_, was employed as the inlet gas to evaluate the sealing properties of the gasket material to comply with the safety guidelines.

## 3. Results and Discussion

The mechanical durability of the elastomers used in this study was tested in terms of the tension and CS behaviors. The TPV materials and EPDM samples used as references were exposed to different aging temperatures, and all TPV and EPDM samples exhibited a decrease in the tensile strength at break with the increase in aging time, as shown in Figure 1a,b, resulting in the embrittlement of the samples, which is mainly attributed to the degradation of molecular chains and the destruction of crosslinking networks [24]. With the increase in aging temperature, the tensile strength at break of EPDM rapidly decreased, as shown in Figure 1a. In contrast, in TPV samples under thermo-oxidative aging, with the increase in aging temperature, the tensile strength slightly decreased and exhibited a fluctuating trend with the increase in aging time; here, the decrease in tensile strength was not significantly larger than that in EPDM, as observed in Figure 1b. These results can be explained by the general feature of TPV, which is well known as a finely dispersed rubber phase, such as EPDM in TPVs. In this case, PP, which has higher crystallinity and a more entangled feature related to thermo-stable properties than general rubber, such as hydrocarbon elastomer, was employed as the matrix in TPV; the matrix material properties were mainly affected within the compounded material when exposed to a high-temperature environment. This occurred because EPDM does not provide greater resistance to breaking than PP under thermal oxidation conditions. Polymer oxidation, which results in chain scissions, was the main factor for the loss of tensile strength. It can be concluded that fewer entanglements of EPDM lead to more rapid reduction in tensile strength with thermal oxidation aging than in the case of PP-based TPV, and a similar trend has been previously reported [25]. Consequently, during the high-temperature aging test, TPV exhibited enhanced sustainable tensile mechanical properties than EPDM due to the presence of entangled thermoplastic PP matrix.

Additionally, the CS property of each material was evaluated under the same condition to define its sealing property based on its elastomer characteristic. As shown in Figure 2a,b, the aging behavior of each material produced widely different curves with the increase in time and temperature. In EPDM, the CS value linearly increased with the increase in temperature and exposure time. However, TPV evaluated under similar conditions exhibited a rapid loss of elastic property compared with EPDM over exposure times shorter than 500 h. Subsequently, the change in CS in TPV saturated, whereas EPDM showed a continuous loss of elasticity above 500 h. For instance, in the case of EPDM, the increase in the ratio of CS over 2000 h at 120 °C was 245% compared with the result over 500 h. However, with the same test procedure, TPV exhibited a ratio below 50% under every test condition. In addition, similar results have been previously reported for incorporated thermoplastic (PP) and elastomer (EPDM) [26]. Therefore, based on these results, we can conclude that the TPV used in this study exhibited more stable mechanical aging properties in terms of the degradation change ratio than elastic EPDM materials in a specific controlled compression process.

Furthermore, the barrier properties of polymeric materials are regarded as the most important indicator of the sealing property. Because the crosslinking density of a thermoset material is proportional to its sealing capability, its crosslinking density should be evaluated under various conditions to compare the sealing properties against liquid or gas penetration into polymeric materials.

In this study, to evaluate the durability of a sealing material in a fuel cell environment, the crosslinking density was investigated using the Flory–Rehner equation as follows:(2)$$V_{e}=\frac{-[\ln(1-V_{r})+V_{r}+\chi V_{r}{}^{2}]}{\left[V1(V_{r}{}^{1/3}-V_{r}/2)\right]}$$
where *V_e_* represents the crosslinking density, *V_r_* represents the gel volume in the swollen sample, *V*_1_ represents the solvent molar volume, and χ represents the polymer–solvent interaction parameter [27]. The gel volume (*V_r_*) in the swollen sample (Equation (2)) was calculated according to the following equation:(3)$$V_{r}=\frac{\frac{m_{p}}{\rho_{p}}}{\frac{m_{p}}{\rho_{p}}+\frac{m_{s}}{\rho_{s}}}$$
where *m_p_* is the weight of the dry polymer, *ρ_p_* is the density of the dry polymer, *m_s_* is the weight of the solvent absorbed by the polymer, and *ρ_s_* is the density of the solvent. The variables used in the above equations are listed in Table 1, and the calculated results are shown in Figure 3a. In this study, the swelling ratio was investigated in two cases, i.e., the pristine material and the CS material exposed to 50 °C.

As shown in Figure 3a, the swelling ratio in the solvent increased with time and attained saturation after 168 h, irrespectively of the tested sample type. In particular, the swelling ratio of the compressed sample with exposure to the operating temperature was slightly higher than that of the pristine material. This indicates a reduction in crosslinking density in the matrix, resulting in the permeation of solvent into the matrix during environmental exposure under specific compression stress. The crosslinking densities calculated using the Flory–Rehner equation are shown in Figure 3b. As shown in Figure 3b, the initial crosslinking density of TPV (1.55381 × 10^−4^ mol cm^−3^) was slightly higher than that of EPDM (1.26192 × 10^−4^ mol cm^−3^). However, despite the reduced crosslinking density, TPV exhibited a larger crosslinking density than EPDM (TPV = 1.28 × 10^−4^ mol cm^−3^; EPDM = 1.00 × 10^−4^ mol cm^−3^). Therefore, it can be concluded that TPV exhibited better barrier properties than EPDM as a sealing material in terms of the crosslinking nature.

After the compression and thermal aging tests, all samples exhibited reduced crosslinking densities compared with that of the pristine material. These results suggest that thermal oxidation and compression stress induced the cleavage of the crosslinked bonding of each elastomer, leading to reduced crosslinking density of the elastomer matrix. It is well known that the EPDM molecular structure undergoes oxidation to afford oxygenated species, such as carbonyl groups, when exposed to a thermo-oxidation environment. Moreover, compressed stresses result in EPDM degradation due to its low degradation activation energy. In this study, the thermal oxidation and degradation of EPDM was also confirmed, in concurrence with the previous literature [30].

The obtained FTIR scan results (Figure 4) show the pristine and compressive stress–thermo-oxidative aging of EPDM and TPV. Overall scan results are illustrated in Figure 4a. The disappearance of the characteristic peak in the fingerprint region at 1100–1200 cm^−1^ is assigned to the sulfur curing agent in both cases [31]. During the thermal oxidation test, these components were volatized, and characteristic peaks also disappeared; similar results have been reported in the recent literature [30,32]. The weakening of the main characteristic peaks of aged EPDM and TPV at 2980 and 718 cm^−1^, respectively, assigned to the –CH_2_- group, provides convincing evidence of the oxidation reaction, as shown in Figure 4b,c.

Furthermore, the aged EPDM samples exhibited a new characteristic peak at 1793 cm^−1^, attributed to the carboxyl carbonyl groups (C=O). The intensity of the peak at 1065 cm^−1^, assigned to the C–O–C group, increased due to the generation of the oxygen group during aging; the degree of oxidation reaction was not sufficient to produce carbonyl compounds, as shown in Figure 4d,e. However, in the case of TPV, characteristic peaks indicating the formation of a carbonyl group were not observed in the FTIR spectra. Therefore, it can be concluded that the thermal oxidation effect of TPV was lower than that of EPDM owing to the less effectively thermally oxidized EPDM volume in the TPV matrix. The results of the FTIR analysis show that the aging degree of EPDM rubber promoted chain scission reactions of EPDM molecules and the generation of oxygenated species. Consequently, the low volume fraction of EPDM in TPV can provide better thermal aging resistance than pristine EPDM because of the strengthened mechanical properties attributed to the PP matrix.

In general, gasket hardness is an important factor governing the seal and waterproof performance of the joint [33]. Ding et al. reported the relationship between gasket hardness and its waterproof performance. According to the previous literature, specifying higher gasket hardness is recommended as a suitable approach to improving waterproof and sealing performance. To investigate the capability of TPV as a fuel cell stack sealing material, the change in the surface hardness under fuel cell operating conditions was investigated in comparison to the corresponding EPDM hardness, and the results are shown in Figure 5a.

As shown in Figure 5a, the initial surface of TPV was much harder than that of EPDM due to incorporation in the PP matrix [34]. These materials were placed in an acidic solution to mimic the fuel cell electrolyte conditions and subjected to ambient temperatures of 70, 80, and 90 °C to investigate the accelerated aging effect compared with the operating temperature (60 °C). With TPV, a hardness value of 76.85 under the no-aging condition was recorded; EPDM exhibited lower hardness (65.9) than TPV under the same condition. With the increase in exposure time and temperature, the hardness of both materials exhibited a similar decreasing behavior, as shown in Figure 5a. This indicates that longer acid exposure times significantly influence the composite hardness values. At acid-aging temperatures of 70, 80, and 90 °C, in TPV, a decrease in the hardness value from 75.04 to 75.07 and 74.99, respectively, was observed. However, these values were still higher than those of EPDM (64.56, 64.43, and 64.21, respectively) obtained under the same conditions. These observations agree with the results by Banna et al. [35], where a similar variation in hardness values was observed. The low strength of EPDM, even while exhibiting crystallinity, resulted from the irregular arrangement of large portions of the rubber chains during initial acid aging, which caused the polymer chains to be loosely bonded by weak van der Waals forces and thus move easily. Therefore, it can be concluded that TPV, which exhibited only a small change in its hardness under acidic conditions, can assure the stability of the clamping force applied on the gas diffusion layer, thus contributing to the stable operation of the fuel cell as a sealing material.

Maintaining the elastic property under operating conditions is the most important factor that contributes to the prevention of the leakage or emission of outer gases (viz., O_2_ and H_2_O). In this study, the CS value of the candidate material and its accelerated aging at high temperatures in an acidic environment were investigated, and the results are illustrated in Figure 5b. As per its definition, CS implies the extent of permanent deformation. Therefore, to investigate the loss of elasticity under operation conditions, the CS data of EPDM and TPV were obtained during prolonged accelerated aging at high temperatures in an acidic environment. The CS values of all materials were found to increase with the increase in time and temperature. Notably, the absolute change in the CS value of TPV was much higher than that of EPDM, regardless of the test temperature. However, the sensitivity of the CS of TPV was below 5% after 500 h of aging, despite its high initial CS value, and it attained saturation after 500 h of aging, as shown in Figure 5b. In comparison, the CS of EPDM exhibited a continuous increase for 1008 h of aging; it also showed greater sensitivity to temperature than TPV based on the investigated CS change, as shown in Figure 5b.

Therefore, it is inferred that TPV exhibits stable properties as a sealing elastomer in a fuel cell environment, such as weak acid electrolyte conditions. In particular, the results of crosslinking density, surface hardness, and deformation of elasticity, compared with EPDM as a reference material, provide clear evidence for the capability of TPV as a gasket material for fuel cell applications.

To demonstrate the surface properties after the aging test under acidic conditions corresponding to the results in Figure 5a,b, the surface morphology of EPDM and TPV was investigated. The surface morphology of the samples exposed to acidic accelerated conditions (diluted H_2_SO_4_ solution (pH 3) at 90 °C for 1000 h) was compared with that of their pristine materials; the results are shown in Figure 6. As shown in Figure 6a,c, the pristine material of TPV showed a significantly rougher and more irregular surface than that of EPDM due to the immiscible phase between EPDM and PP materials in TPV. After the aging procedure, both cases exhibited a similar change in appearance, as shown in Figure 6b,c. EPDM and TPV after the aging test showed weaker swollen surface features than the surface of the pristine materials. This indicates that the acidic component in solution could penetrate the aged sample and led to changes in surface properties that are related to the change in hardness, as illustrated in Figure 5a.

Based on the aforementioned feasibility test, the evaluation of sealing performance at the cell level is necessary to demonstrate the capability of TPV as a sealing gasket. Thus, in this study, a simple stack leakage test was designed to evaluate the sealing effect of the gasket material. The test design is shown in Figure 7, wherein four layers of the gasket material and the metal frame were stacked alternately and then compressed to a 40% compression ratio to obtain a reasonable sealing effect [36]. After assembling, He gas was employed to evaluate the quantity of leakage from the stacked cell. He gas was injected using a pressure control regulator to maintain a constant injection pressure, and a thermal cycle test was conducted before the leakage test to introduce internal stresses between the frame and the gasket material owing to the difference in thermal expansion behavior. The thermal cycle test profile data are listed in Table 2. Note that the pressure of injected He was held constant during examination and was varied using 0.2-bar increments from 0 to 1.0 bar.

Figure 8 shows the leakage rates of the EPDM and TPV gasket materials at different test temperatures (−35, 25, and 95 °C) to investigate the operation capability of the gasket in an automotive application environment. Additionally, the inlet He pressure into the stacked cell was controlled using 0.2-bar increments to provide the trend of gasket leakage under various power generation conditions of the stacked cell. The leakage rates were calculated using the below equation, and the results are shown in Figure 8.
(4)$$L=\frac{\Delta PV_{e}}{P_{0}\Delta tC}$$
where *L* is the leakage rate (sccm·cm^−1^ (standard cubic centimeter per minute per centimeter)), Δ*P* is the pressure difference (kPa), *P*_0_ is the standard atmospheric pressure (kPa), Δ*t* is the leakage time (min), *V_e_* is the equivalent internal volume (cm^3^), and *C* is the outer length of the seal circumference (cm).

As shown in Figure 8, the leakage rates of all gasket materials increased upon the increase in the inlet gas pressure from 0.2 to 1 bar. Similar results were observed with the increase in temperature. The results indicate that the maximum leakage rates of EPDM and TPV were 1.25 × 10^−6^ and 1.23 × 10^−6^ sscm·cm^−1^, respectively, at the inlet pressure of 1 bar and the operating temperature of 95 °C. The difference in the leakage performance of TPV and EPDM as commercialized gasket materials in automotive PEM fuel cell applications was about 2.05 × 10^−8^ sscm·cm^−1^, 1% below the offset compared with the total leakage rate. Therefore, it is reasonable to conclude that the investigation of the leakage rates of these materials results in similar expected capability as a sealing gasket material regardless of the operating conditions.

Further, to analyze the sealing performance of TPV, the results of this study were compared with those in the recently reported literature. Wang et al. investigated the sealing performance of EPDM and silicon as gasket materials in PEMFC, and they revealed EPDM leakage rates of 1.1 × 10^−7^ and 1.1 × 10^−6^ Pa·m^3^·s^−1^ at 40% compression, which were caused by interfacial leak and the gas permeation effect, respectively [36]. To effectively compare the results obtained in the literature and those of this study, relevant unit conversions were applied (1 Pa·m^3^·s^−1^ = 600 sccm; gasket circumference in this study = 80 cm), and the results are shown in Figure 7. The results indicate that the leakage rate calculated in this study is located within the range of silicon and EPDM leakage rates obtained in the literature. Despite the existence of a few different control factors, such as interfacial condition and clamping force, the comparison of the leakage rates indicates clear differences among silicon, TPV, and EPDM used in the current and previous studies. TPV and EPDM exhibited similar performance within the range from 7.86 × 10^−7^ to 1.25 × 10^−6^ sscm·cm^−1^, with a leakage rate difference of 4.65 × 10^−7^ sscm·cm^−1^. However, the silicon gasket (8.25 × 10^−6^ sscm·cm^−1^) exhibited seven times higher leakage than TPV (1.23 × 10^−6^ sscm·cm^−1^) and EPDM (1.25 × 10^−6^ sscm·cm^−1^).

Thus, the results indicate that TPV exhibits sealing performance similar to that of EPDM, which is already adopted as a gasket material in automotive applications, while exhibiting enhanced performance compared with the reference material in terms of mechanical stability in a high-temperature and acidic environment.

## 4. Conclusions

In this study, TPV composed of crosslinked EPDM in PP matrix was evaluated as a promising PEM gasket material to substitute EPDM gaskets. The long-term stability of the mechanical properties, crosslinking density, and property changes of TPV and EPDM under acidic conditions were evaluated and compared to determine the advantages of TPV as a gasket sealing material.

The long-term mechanical stability of TPV was evaluated by testing the tension and CS under accelerated aging conditions. Compared with EPDM, TPV exhibited enhanced sustainable tensile mechanical properties and a stable degradation change ratio in a specific controlled compression process under high-temperature aging conditions.

Further, the change in crosslinking density with the increase in compression stress and temperature was investigated to confirm the change in molecular structure necessary for maintaining the elastic property, which can provide effective elastic sealing performance. A higher crosslinking density of TPV was calculated based on the Flory–Rehner theory and compared with that of EPDM. The rationale behind these results was confirmed using FTIR spectroscopy. The aged EPDM exhibited newly generated carboxyl carbonyl groups (C=O, –C–O–C–), whereas the peaks assigned to –CH_2_ groups weakened, providing convincing evidence of oxidation reaction. Therefore, EPDM oxidation led to its reduced elastic property, which justifies the decreased crosslinking density and cleavage of the molecular network.

To investigate the expected sealing properties, crosslinking density and surface hardness were also evaluated under accelerated temperature conditions. The measured surface hardness of TPV was much higher than that of EPDM regardless of the test temperature and aging time.

Finally, the sealing performance of each material was confirmed with a simple stack leakage test. The results show similar leakage rates for TPV and EPDM for a range of test inlet pressure values of He regardless of the applied temperature. Thus, it can be concluded that TPV exhibits a similar sealing capability as a gasket material in terms of He leakage performance.

Therefore, the TPV investigated in this study can be considered a candidate gasket material with similar sealing performance and more stable mechanical properties compared with EPDM, which has been adopted in PEM fuel cell applications as a sealing gasket material.

## Figures and Tables

**Figure 1 polymers-15-01872-f001:**
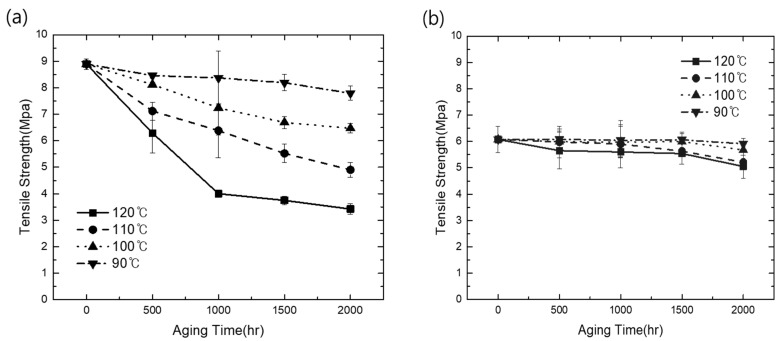
Comparison of tensile thermal oxidation aging results: (**a**) tensile strength changes of EPDM; (**b**) tensile strength changes of TPV.

**Figure 2 polymers-15-01872-f002:**
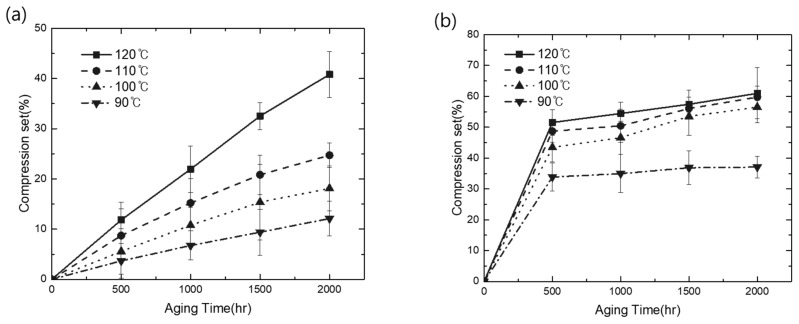
Comparison of CS thermal oxidation aging results: (**a**) CS changes of EPDM; (**b**) CS changes of TPV.

**Figure 3 polymers-15-01872-f003:**
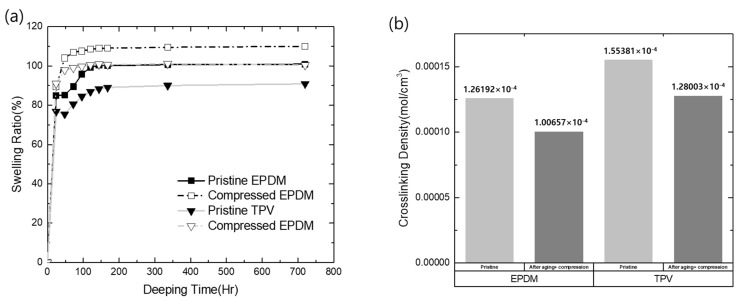
The crosslinking properties of EPDM and TPV: (**a**) swelling ratio changes of EPDM and TPV in cyclohexane; (**b**) calculated crosslinking density values of EPDM and TPV.

**Figure 4 polymers-15-01872-f004:**
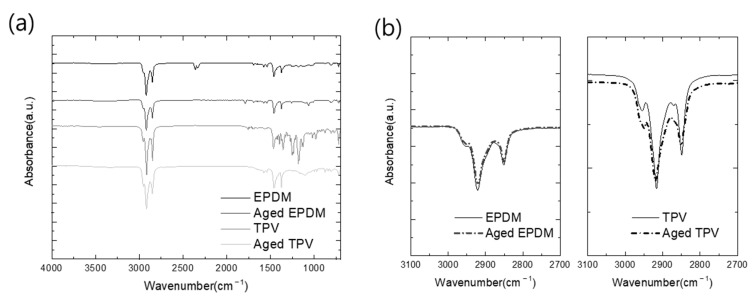
Comparison of FTIR spectra of pristine and thermal oxidized EPDM: (**a**) full-range scan results; (**b**,**c**) peaks assigned to CH_2_ stretching and vibration; (**d**,**e**) peaks assigned to carbonyl stretching and vibration.

**Figure 5 polymers-15-01872-f005:**
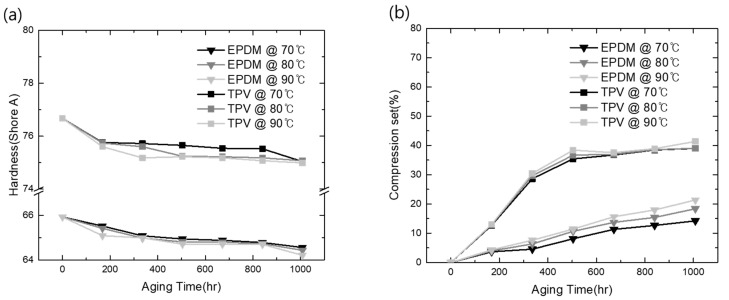
Mechanical property changes under acid conditions: (**a**) surface hardness changes; (**b**) CS changes.

**Figure 6 polymers-15-01872-f006:**
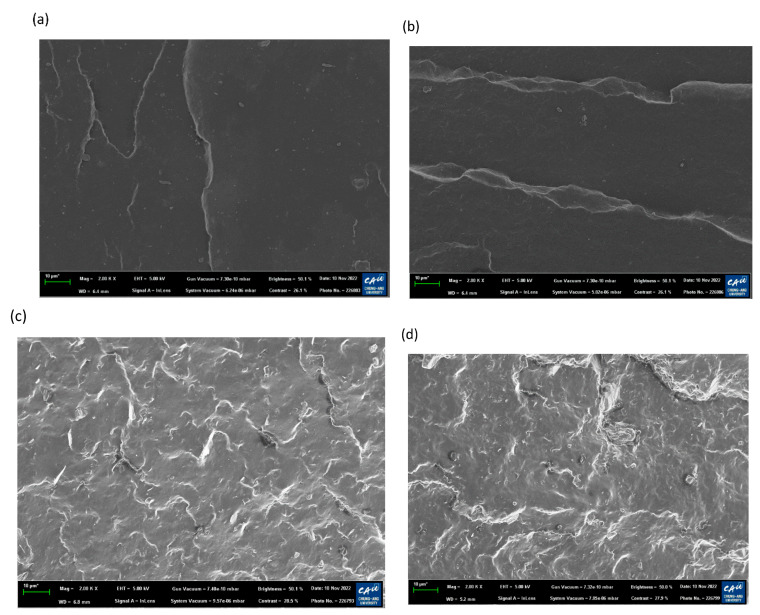
Surface photomicrographs of pristine and aged materials under 90 °C acidic condition; (**a**) pristine EPDM, (**b**) aged EPDM, (**c**) pristine TPV, and (**d**) aged TPV.

**Figure 7 polymers-15-01872-f007:**
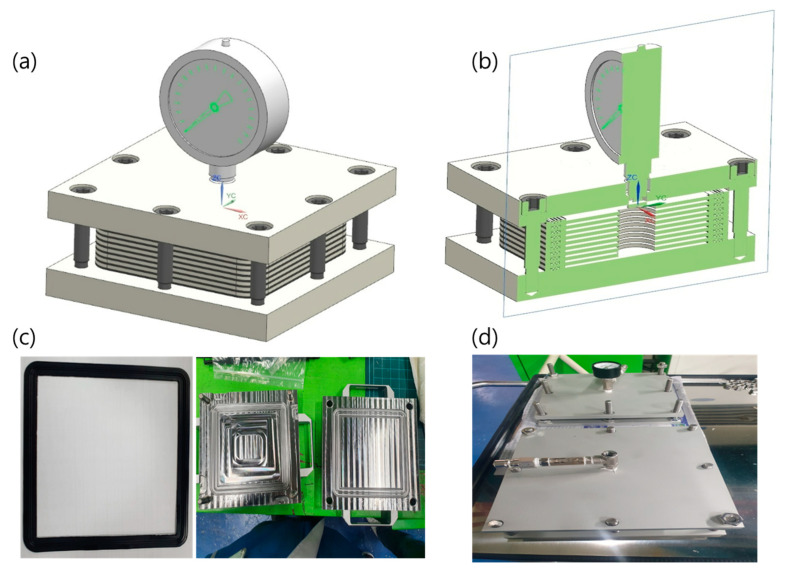
Experimental fixture for evaluation of leakage performance in gasket stack: (**a**) schematic diagram of overall fixture; (**b**) cross-section schematic diagram of fixture; (**c**) image of gasket and sealing plate; (**d**) image of assembled test fixture.

**Figure 8 polymers-15-01872-f008:**
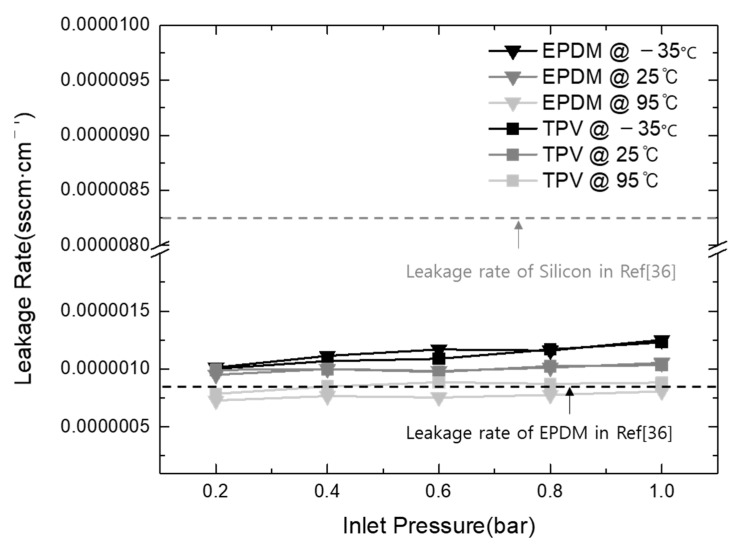
Comparison of leakage rates of gasket materials prepared in this study with results from the literature.

**Table 1 polymers-15-01872-t001:** Variable parameters of Flory–Rehner equation used in this study.

	W_d_	W_s_	α	ρ	ρ_s_	Χ [28,29]	V_0_
**EPDM**	1.5711	3.2913	0.477349	0.86	0.774	0.321	108.7
**TPV**	1.4973	3.0264	0.494746	1.12	0.774	0.321	108.7

**Table 2 polymers-15-01872-t002:** Thermal cycle profile applied to leakage test cell used in this study (repeated 10 times).

	1st Step	2nd Step	3rd Step	4th Step	5th Step
**Temperature (°C)**	25	-	150	-	−45
**Duration time (min)**	-	-	30	-	30
**Ramping rate (°C/min)**	-	10	-	−10	-

## Data Availability

The data presented in this study are available on request from the corresponding author.

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
