# Peer review of "Mechanical Aging Test and Sealing Performance of Thermoplastic Vulcanizate as Sealing Gasket in Automotive Fuel Cell Applications"

_polymers, 2023, doi:10.3390/polym15081872_

Round 1
Reviewer 1 Report
This study is very interesting and this study offers a good alternative gasket for fuel cells.
Few comments to further improve the quality of this manuscript
1. Can you please provide FTIR spectrum of TPV in addition to EPDM so you can compare and contrast. This will add further proof to your conclusion
2. Can you please explain the mechanism by which the compression set change for TPV saturates at 500 hr of ageing in Fig 2.
3. Have you considered polymers other than PP to make the TPV?
Author Response
I attached response about reviewer1's comments.
Best Regards

Reviewer 2 Report
The manuscript reported TPV as a gasket material for PEM fuel cell applications. It is found that TPV showed better long-term stability in terms of tensile and compression set behaviours under accelerated ageing conditions than EPDM. Additionally, TPV exhibited significantly higher crosslinking density and surface hardness than EPDM regardless of the test temperature and ageing time. TPV and EPDM showed similar leakage rates for the entire range of test inlet pressure regardless of the applied temperature. It can be concluded that TPV exhibits a similar sealing capability with more stable mechanical properties compared to a commercialized EPDM gasket.
The content of this manuscript meets the reading interests of the readers of the journal. However, there are certain English spelling and grammar issues, and also the discussion and explanation should be further improved. I suggest giving a minor revision and the authors need to clarify some issues or supply some more experimental data to enrich the content.
1. For grammar issues, it is suggested that the author double-check the small grammar errors in the full text, especially the lack of and redundant use of definite articles.
2.For the Keywords, 'mechanical ageing', 'sealing performance', 'fuel cell', 'thermoplastic vulcanizate', and 'EPDM/TPV' should be added in order to attract a broader readership.
3. Page 1, 'Hydrogen energy, a clean and flexible secondary energy, is considered the preferred choice for long-range transport, energy-intensive industrial processes, and supporting the integration of decentralized renewable energy, such as wind and solar energy [1-5].' The relationship between hydrogen and renewable energy seems not very clear. I suggest making slightly longer descriptions for this part. For example, the unique intermittent and instability of renewable energy have brought major challenges to the stable operation of the power system, opening temporal and spatial gaps between the consumption of the energy by end-users and its availability, thus, electrochemical energy storage technology is an effective means that can help achieve stable and efficient renewable energy (Batteries 2022, 8(11), 202).
4. Page 2, 'However, there have been few reported studies on the use of TPV as a fuel cell gasket material based on its electro-chemical resistance properties despite its good processability and cost merits.' It is not very clear what is the meaning of 'electrochemical resistance properties'. Judging by the sentence, it seems a disadvantage. However, if it resists electrochemical reactions and keeps stable, it should be an advantage. Otherwise, if it leads to more ohmic loss, it should be 'electrical resistance' rather than 'electrochemical resistance'. This issue should be well-addressed.
5. Page 5, 'However, TPV evaluated under similar conditions exhibits a rapid loss of elastic property compared to EPDM for exposure times below 500 h, and less change ratio of its property than the result of EPDM above 500 h. ' The description of this sentence seems to be inconsistent. It seems for loss of elastic property, TPV behaves worse than EPDM. While at the same time, it has less change of property compared to EPDM. I consider the loss of elastic property to be also the change of property. So it is very hard to understand these sentences.
6. Page 7, 'In this study, the swelling ratio was investigated for two cases, i.e. the pristine material and the compression set material exposed to 50 °C, which imitates the condition of the fuel cell operating temperature.' However, it should be noted most PEMFCs operate around 60 to 85 °C (Fuels 2022, 3(3), 449-474). It seems 50 °C is a bit low for the operation of PEMFC. Hence, the selection of such a temperature needs more reasons.
7. Page 13, 'These materials were placed in an acidic solution to mimic the fuel cell electrolyte condition and subjected to ambient temperatures of 70, 80, and 90 °C to investigate the accelerated ageing effect compared with the operating temperature (50 °C).' What is the composition of the acidic solution exactly to mimic the fuel cell electrolyte condition? It should be provided. And if the operating temperature is just 50 °C, using ambient temperatures may really accelerate the ageing effect. But I am afraid these ambient temperatures should be the correct operating temperature. Hence, the accelerating effect may not be realized by the current ambient temperatures.
8. Page 13, 'Maintaining the elastic property under operating conditions is the most important factor that contributes to the prevention of leakage of the internal components (viz. electrolyte) and penetration of outer gases (viz. O2 and H2O). ' The leakage of the electrolyte is very strange. The electrolyte should be the polymer electrolyte membrane (such as Nafion family membranes), which is solid and no leakage should take place. The mechanical and chemical stability is rather good. And the penetration of outer gases, it seems takes place through the membrane for crossover/penetration, hence the Nafion membrane should be an efficient barrier layer to prevent crossover issues (Electrochimica Acta 378 (2021): 138133.). Through the gasket, it should be the leakage or emission of outer gases, not penetration.
9. Page 14, 'Figure 6. Surface photomicrographs of pristine and aged material in 90℃ acidic conditions; (a) Pristine EPDM, (b) Aged EPDM, (c) Pristine TPV and (d) Aged EPDM.' I consider (d) should be 'aged TPV', rather than 'aged EPDM', since (b) and (d) are exactly the same.
Author Response
I attached response about reviewer 2's comments.
Best Regards

Reviewer 3 Report
This manuscript investigated the mechanical property changes (such as tensile and compression set behaviors, hardness change, and physical free volume change, etal) of TPV materials under accelerated aging conditions, and explored the promising applications as a gasket material in automotive PEM fuel cell. The manuscript is clear and logical for reading. It is suggested to be published after a minor revision. Whether there is a difference in vulcanized curves and rheological properties between EPDM and TPV? The curves in Figure 1 and 2 is hard to observe clearly, it is suggested to use different color to distinguish samples.
Author Response
I attached response about reviewer3's comments.
Best Regards
